# Glucokinase activity controls peripherally located subpopulations of β-cells that lead islet Ca²⁺ oscillations

**Erli Jin[1†], Jennifer K Briggs[2†], Richard KP Benninger[2,3]\*, Matthew J Merrins[1]\***

[1]Department of Medicine, Division of Endocrinology, Diabetes & Metabolism, University of Wisconsin-Madison, Madison, United States; [2]Department of Bioengineering, University of Colorado Anschutz Medical Campus, Aurora, United States; [3]Barbara Davis Center for Childhood Diabetes, University of Colorado Anschutz Medical Campus, Aurora, United States

## eLife Assessment

This study provides **compelling** evidence for functional subpopulations of β-cells responsible for Ca2+ signal initiation and maintenance using novel three-dimensional light sheet microscopy imaging and analysis of pancreatic islets. The findings are **important** as they help decode mechanistic underpinnings of islet calcium oscillations and the resulting pulsatile insulin secretion. The work will be of general interest to cell biologists and particular interest to islet biologists.

**\*For correspondence:**
richard.benninger@cuanschutz.
edu (RKPB);
merrins@wisc.edu (MJM)

[†]These authors contributed
equally to this work

**Competing interest:** The authors declare that no competing interests exist.

**Abstract** Oscillations in insulin secretion, driven by islet Ca²⁺ waves, are crucial for glycemic control. Prior studies, performed with single-plane imaging, suggest that subpopulations of electrically coupled β-cells have privileged roles in leading and coordinating the propagation of Ca²⁺ waves. Here, we used three-dimensional (3D) light-sheet imaging to analyze the location and Ca²⁺ activity of single β-cells within the entire islet at >2 Hz. In contrast with single-plane studies, 3D network analysis indicates that the most highly synchronized β-cells are located at the islet center, and remain regionally but not cellularly stable between oscillations. This subpopulation, which includes 'hub cells', is insensitive to changes in fuel metabolism induced by glucokinase and pyruvate kinase activation. β-Cells that initiate the Ca²⁺ wave (leaders) are located at the islet periphery, and strikingly, change their identity over time via rotations in the wave axis. Glucokinase activation, which increased oscillation period, reinforced leader cells and stabilized the wave axis. Pyruvate kinase activation, despite increasing oscillation frequency, had no effect on leader cells, indicating the wave origin is patterned by fuel input. These findings emphasize the stochastic nature of the β-cell subpopulations that control Ca²⁺ oscillations and identify a role for glucokinase in spatially patterning 'leader' β-cells.

## Introduction

Pulsatile insulin secretion from pancreatic islet β-cells is key to maintaining glycemic control. Insulin secretory oscillations increase the efficiency of hepatic insulin signaling and are disrupted in individuals with obesity and diabetes (*Satin et al., 2015*). The primary stimulus for insulin release is glucose, which is intracellularly metabolized to generate a rise in ATP/ADP ratio, which closes ATP-sensitive K⁺ channels (K$_{ATP}$ channels) to initiate Ca²⁺ influx and insulin secretion (*Merrins et al., 2022*). At elevated glucose, β-cells oscillate between electrically silent and electrically active phases with a period of minutes, both in vivo and in isolated islets. The depolarizing current can be transmitted between β-cells

across the whole islet through gap junction channels. However, the activity of individual electrically coupled β-cells is functionally heterogenous (*Benninger and Kravets, 2022*; *Hiriart and Ramirez-Medeles, 1991*; *Kiekens, 1992*; *Pipeleers, 1992*; *Rutter et al., 2024*; *Wojtusciszyn et al., 2008*; *Da Silva Xavier and Rutter, 2020*). This heterogeneity results in the emergence of β-cell subpopulations that may be crucial for maintaining the coordination of the whole islet and regulating pulsatile insulin release (*Johnston et al., 2016*; *Kravets et al., 2022*; *Salem et al., 2019*; *Westacott et al., 2017*). Understanding the underpinnings of β-cell functional heterogeneity and islet cell communication is important for understanding islet dysfunction and the pathogenesis of diabetes.

Similar to studies of neuronal networks, functional network analysis can be used to quantify interactions within the heterogenous β-cell system. Interactions (termed 'edges') are drawn between β-cell pairs with highly correlated $Ca^{2+}$ dynamics. Studies suggest that the β-cell functional network exhibits high clustering or 'small-world' properties (*Stožer et al., 2013b*), with a subpopulation of β-cells that are highly synchronized to other cells (hub cells) (*Johnston et al., 2016*). Silencing the electrical activity of these hub cells with optogenetics was found to abolish the coordination within that plane of the islet (*Johnston et al., 2016*; *Dwulet et al., 2021*; *Nasteska et al., 2021*). Similarly, time series-based lagged cross correlation analysis has identified subpopulations of cells at the wave origin, termed 'early phase' or 'leader cells', that lead the second-phase $Ca^{2+}$ wave by depolarizing and repolarizing first (*Salem et al., 2019*; *Westacott et al., 2017*). However, questions have been raised whether the highly networked or leader subpopulations have the power to control the entire islet (*Dwulet et al., 2021*; *Briggs et al., 2023*; *Peercy and Sherman, 2022*; *Satin et al., 2020*). Underlying this controversy lies several unanswered questions: what mechanisms drive the existence of these functional subpopulations? Do these subpopulations arise primarily from mechanisms intrinsic to β-cells, making the subpopulations consistent over time? Alternatively, do they arise from the combination of intrinsic mechanisms and emergence due to surrounding cells, allowing the subpopulations to fluidly change over time? To date, experiments have been restricted to imaging a single two-dimensional (2D) plane of the islet which contains only a small fraction of the β-cells present in the three-dimensional (3D) islet tissue, limiting the ability to address these questions.

With these caveats in mind, prior studies using a mixture of computational and molecular approaches suggested that β-cell subpopulations are patterned by glucokinase, which is often referred to as the 'glucose sensor' for the β-cell (*Westacott et al., 2017*; *Dwulet et al., 2019*; *Jetton and Magnuson, 1992*; *Briggs et al., 2023*). By phosphorylating glucose in the first step of glycolysis, glucokinase activation lengthens the active phase of $Ca^{2+}$ oscillations by committing more glucose carbons to glycolysis (*Lewandowski et al., 2020*). Until recently, it was believed that downstream glycolysis was irrelevant to pulsatile insulin secretion. However, in conflict with this model, allosteric activation of pyruvate kinase accelerates $Ca^{2+}$ oscillations and increases insulin secretion (*Lewandowski et al., 2020*; *Foster et al., 2022*). As a potential mechanistic explanation for these observations, plasma membrane-associated glycolytic enzymes, including glucokinase and pyruvate kinase, have been demonstrated to regulate $K_{ATP}$ channels via the ATP/ADP ratio (*Ho et al., 2023*). However, it remains unknown whether these glycolytic enzymes influence β-cell heterogeneity and network activity.

To study single β-cell activity within intact islets, we engineered a 3D light-sheet microscope to simultaneously record the location and $Ca^{2+}$ activity of single β-cells over the entire islet during glucose-stimulated oscillations. In concert, we developed 3D analyses to investigate the spatial features of subpopulations that underlie the β-cell network and $Ca^{2+}$ wave, and the consistency of these features over time. We further examined the consequences of sampling islet heterogeneity in 2D compared to 3D. Finally, we investigated the role of the glycolytic enzymes glucokinase and pyruvate kinase in controlling β-cell subpopulations during glucose-stimulated oscillations.

## Results

### Light-sheet microscopy enables high-speed 3D imaging of oscillations in single β-cells within intact islets

To acquire high-speed 3D time course imaging of β-cell $Ca^{2+}$ oscillations within intact islets, we utilized a lateral-interference tilted excitation light-sheet system (*Fadero et al., 2018*) mounted on an inverted fluorescence microscope (*Figure 1A* and *Methods*). To image $Ca^{2+}$ activity and spatially resolve individual β-cells in intact islets, islets were isolated from *Ins1-Cre:ROSA26^GCaMP6s/H2B-mCherry* mice

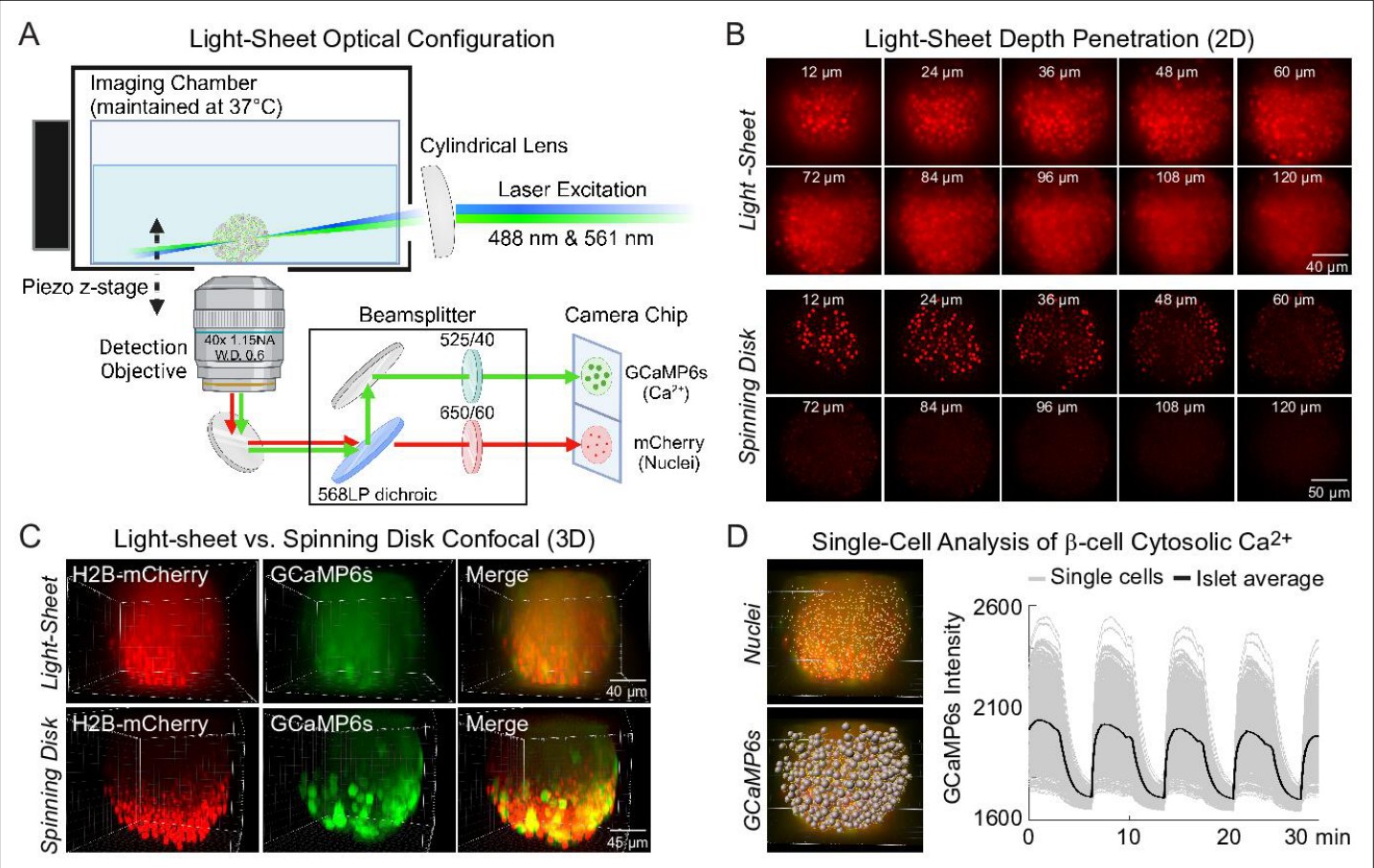

**Figure 1.** Engineering of a light-sheet microscope to image intact islets in 3D. (**A**) Schematic of the light-sheet microscope showing the optical configuration. (**B**) Representative light-sheet (upper panel) and spinning disk confocal images (lower panel) of a mouse pancreatic islet expressing β-cell-specific H2B-mCherry fluorophore at different two-dimensional (2D) focal planes, emphasizing the superior depth penetration of the light-sheet microscope. (**C**) 3D imaging of β-cells expressing GCaMP6s Ca²⁺ biosensors and nuclei mCherry biosensors. (**D**) Using *Ins1-Cre:ROSA26^GCaMP6s/H2B-mCherry* islets, the software-identified center of β-cell nuclei (yellow dots) was used to generate GCaMP6s regions of interest (gray spheres). A representative Ca²⁺ time course is displayed in the right panel for an islet stimulated with glucose and amino acids.

The online version of this article includes the following figure supplement(s) for figure 1:

**Figure supplement 1.** Hardware wiring diagram of the light-sheet microscope.

**Figure supplement 2.** NIS-Elements software configuration.

that express cytosolic GCaMP6s Ca²⁺ biosensors and nuclear H2B-mCherry reporters selectively within β-cells. We first compared the images collected by the light-sheet system with a commercial spinning disk confocal using the same ×40 water immersion objective. Similar to a widefield microscope, the axial resolution of the light-sheet microscope is dictated by the numerical aperture (NA) of the objective lens (~1.1 μm for a 1.15 NA objective and GCaMP6s emission) (**Fadero et al., 2018**), whereas the spinning disk uses a pinhole array to enhance axial resolution. At a shallow depth of 24 μm from the coverslip, H2B-mCherry-labeled nuclei and GCaMP6s-labeled β-cells were resolved both by the light-sheet and the spinning disk confocal. However, the nuclei were only resolved by the light-sheet system at depths ≥60 μm due to the reduced light scatter from side illumination (**Figure 1B**). Thus, the main advantage of the light-sheet system is the ability to image the entire islet in 3D (**Figure 1C**).

Prior studies of β-cell Ca²⁺ oscillations utilized 1 Hz imaging to resolve phase shifts for single β-cell traces within a single 2D plane (**Benninger et al., 2008**; **Hraha et al., 2014**; **Skyggebjerg, 1999**; **Stožer et al., 2013a**). To image the entire islet at similar acquisition speeds, the hardware was operated under triggering mode to minimize communication delays (**Figure 1—figure supplement 1**). In this mode, the lasers and the piezo z-stage were triggered directly by the camera, which received a single set of instructions from the computer via the NiDAQ card. To image 132 μm into the islet at

2 Hz, 15 ms was allowed for photon collection and stage movement for each of the 4 μm z-steps. Initially, an hour-long delay was required to save 120,000 imaging files after running a continuous 30-min experiment. Because this delay is only observed after the first 3 min of imaging, it was possible to eliminate the delay by separating the acquisition into a series of 3-min loops (*Figure 1—figure supplement 2*).

Individual β-cell nuclei were located using the Spots function of Bitplane Imaris software (*Figure 1D*). For each nucleus, a cellular region of interest (ROI) was defined by a sphere of radius 4.65 μm around the ROI center based on results from a computational automated radius detection (*Briggs et al., 2024*). The mean GCaMP6s intensity of each cell ROI for each time point was calculated and exported as single-cell traces. We did not observe any significant photobleaching using continuous GCaMP6s and H2B-mCherry excitation over the course of the experiment. The combination of this light-sheet system, islet cell labeling, and analysis pipeline allows for imaging of $Ca^{2+}$ from nearly all β-cells in the islet at speeds fast enough for spatio-temporal analyses to identify functionally heterogenous β-cell subpopulations.

## 3D analyses of islet $Ca^{2+}$ oscillations reveal that the β-cell network is distributed in a radial pattern while $Ca^{2+}$ waves begin and end on the islet periphery

To investigate the synchronization between β-cells across the islet in 3D space, we imaged and extracted $Ca^{2+}$ time courses for *Ins1-Cre:ROSA26$^{GCaMP6s/H2B-mCherry}$* islets that exhibit slow oscillations. Following the network analysis methods set forth in *Šterk et al., 2024*, we calculated the correlation coefficient between every cell pair and defined an 'edge' between any cell pairs whose correlation coefficient was above threshold (*Figure 2A*). This threshold was set such that the average number of edges per cell, also called the 'cell degree', was equal to 7. A fixed average degree rather than fixed threshold was used to mitigate inter-islet heterogeneity (*Šterk et al., 2024*). An example 3D network for a single β-cell within an islet is shown (*Figure 2B*) along with the frequency distribution of all β-cells within the islet (*Figure 2C*). The high degree cells (top 10% of the total population, *blue*) and low degree cells (bottom 10% of the total population, *red*) were then mapped onto a 3D projection of the islet and onto the $Ca^{2+}$ time course (*Figure 2D*). Compared to average degree cells, the high degree cells were consistently located at the center of the islet while the low degree cells were located on the periphery, indicating that the islet network is distributed in a radial pattern (*Figure 2D, E*).

To analyze the propagation and spatial orientation of $Ca^{2+}$ wave in 3D space, we calculated the lagged correlation coefficient between every β-cell and the islet average, and identified the phase lag with maximum correlation (*Figure 2F*). The spatial distribution of phases of an example islet is shown (*Figure 2G*), along with the frequency distribution of all β-cells within the islet (*Figure 2H*). The early phase cells (top 10% of the total population that depolarize first and repolarize first, *blue*) and late phase cells (bottom 10% of the total population that depolarize last and repolarize last, *red*) were then mapped onto a 3D projection of the islet and onto the $Ca^{2+}$ time course (*Figure 2I*). Unlike the islet network, for which the high degree cells emanate from the islet center (*Figure 2D, E*), the early and late phase cells were each located at the islet periphery, and show a clear temporal separation between depolarization and repolarization (*Figure 2I, J*).

## The location of the β-cell network is stable over time while the wave progression varies

We next assessed the stability of high degree cells and early phase cells over time, by assessing their presence across consecutive oscillations (*Figure 3A, B*; *Figure 3—figure supplement 1*). The high degree cells and early phase cells have a similar ~60% retention rate between oscillations (*Figure 3C*). Strikingly, when we examined the center of gravity for each β-cell subpopulation, we found that the center of gravity of the early phase cells moved significantly more than that of the high degree cells (*Figure 3D*). This indicates that the early phase cells tend to change their identity more with each oscillation. To further investigate the change in location of early phase cells, we used principal component analysis to identify the principal axis between early and late phase cells (wave axis) and calculated the rotation of the axis between each oscillation. Of the 25 islets examined, 57% show substantial changes in the wave axis over time (*Figure 3E*). Thus, β-cell depolarization is initiated at different locations within the islet over time, while the β-cell network location is relatively stable.

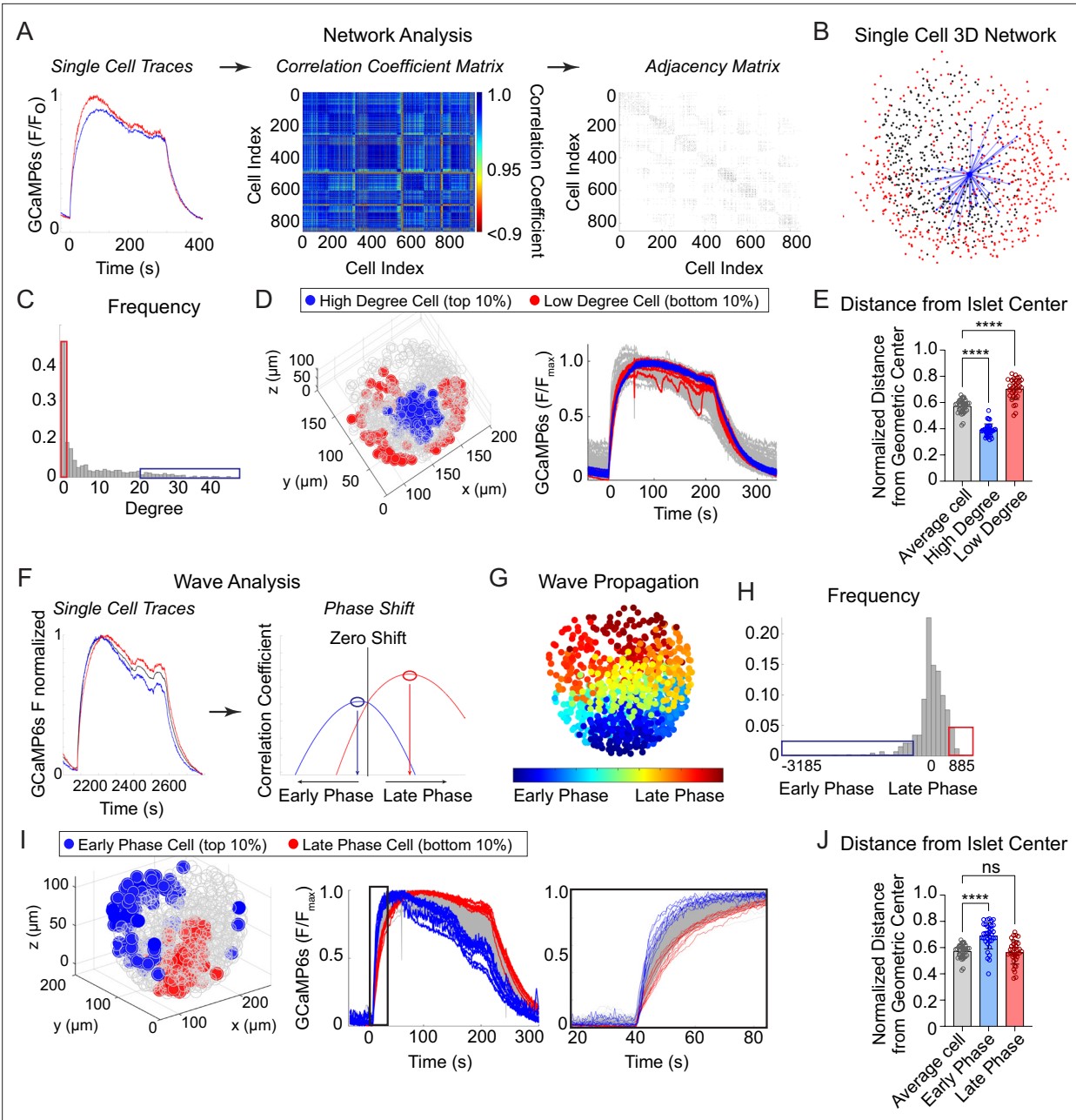

**Figure 2.** Characterization of single β-cells using three-dimensional (3D) network and phase analysis. (**A**) Flow diagram illustrating the calculation of cell degree from pairwise comparisons between single β-cells. (**B**) An example 3D network for a single β-cell within a representative islet is shown with synchronized cell pairs in blue, cells that have other synchronized pairs in black, and cells that are asynchronous in red. This analysis is repeated for all cells in the islet. (**C**) Frequency distribution of cell degree for all β-cells analyzed. Top 10% (blue box) and bottom 10% (red box) are high and low degree cells. (**D**) Representative 3D illustration and $Ca^{2+}$ traces showing the location of high degree cells (blue) and low degree cells (red). (**E**) Quantification of the normalized distance from the islet center for average degree cells (gray), high degree cells (blue), and low degree cells (red). (**F**) Flow diagram illustrating the calculation of cell phase, calculated from the correlation coefficient and phase shift. (**G**) Wave propagation from early phase cells (blue) to late phase cells (red) in 3D space. (**H**) Frequency distribution of cell phase for all β-cells analyzed. Top 10% (blue box) and bottom 10% (red box) are early and late phase cells. (**I**) Representative 3D illustration and $Ca^{2+}$ traces showing the location and traces of high phase cells (blue) and low degree cells (red). (**J**) Quantification of the normalized distance from the islet center for average phase cells (gray), early phase cells (blue), and late phase cells (red). Data represent n = 28,855 cells, 33 islets, 7 mice. Data are displayed as mean ± SEM. ****p < 0.0001 by one-way ANOVA.

The online version of this article includes the following source data for figure 2:

**Source data 1.** Source data for distance from center measurement for network and wave analysis.

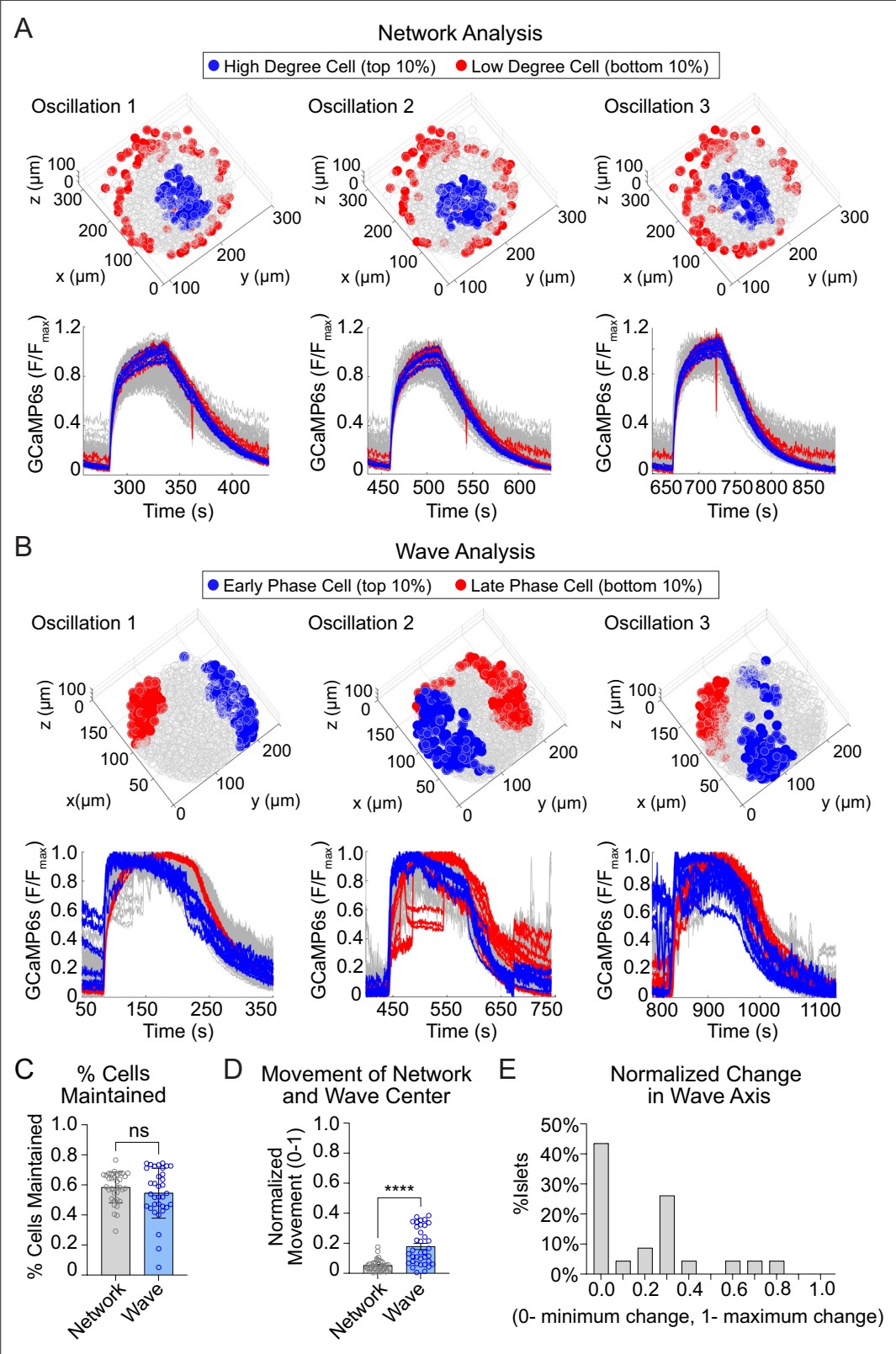

**Figure 3.** The network of highly synchronized β-cells is consistent between oscillations, while the Ca²⁺ wave axis rotates. (**A**) Three-dimensional (3D) representation of the islet showing the location of high degree cells (blue) and low degree cells (red) over three consecutive oscillations (top panel) and their corresponding Ca²⁺ traces (bottom panel). (**B**) 3D representation of the islet showing the location of early phase cells (blue) and late phase

*Figure 3 continued on next page*

*Figure 3 continued*

cells (red) over three consecutive oscillations (top panel) and their corresponding Ca²⁺ traces (bottom panel). (C) Quantification of the retention rate of high degree and early phase cells. (D) Relative spatial change in the center of gravity of β-cell network versus the β-cell Ca²⁺ wave. (E) Frequency distribution showing the normalized change in Ca²⁺ wave axis for all islets. Data are displayed as mean ± SEM. ****p < 0.0001 by normality test followed by Paired Student's *t*-test or Wilcoxon Signed-Rank Test.

The online version of this article includes the following source data and figure supplement(s) for figure 3:

**Source data 1.** Source data for % cells maintained, movement of network/wave center and wave axis change analysis.

**Figure supplement 1.** Location of early and late phase cells in an islet with stable wave axis.

The analyses in *Figure 3* are focused on the top and bottom 10% of the population. To understand the stability of all β-cells within the 3D network or the 3D wave propagation over time, we ranked every β-cell in the islet by their phase/degree (cellular consistency), as well as the spatial proximity of every β-cell to the center of gravity of the top 10% of the subpopulation (regional consistency) (*Figure 4A*). We quantified the change in these distributions using a normalized non-parametric, information-theoretic metric termed Kullback–Leibler (KL) divergence (see *Methods*). If the ranking of high to low degree cell (e.g., A > B > C) for the first oscillation remains the same in the second oscillation, the KL divergence will be 0, indicating the cell ranking is completely predictable between oscillations. Alternatively, if the cell ranking changes between oscillations (A > C > B), the KL divergence will be 1, indicating the cell ranking is completely random (*Figure 4B*). When examining the consistency of the network, the regional stability was much higher than the cellular stability over time (*Figure 4C*). In contrast, when examining the wave, the cellular stability was similar to the regional stability (*Figure 4D*). This analysis of KL divergence supports the previous conclusions that the β-cell network is regionally stable, but the wave can start at different locations. Additionally, because the

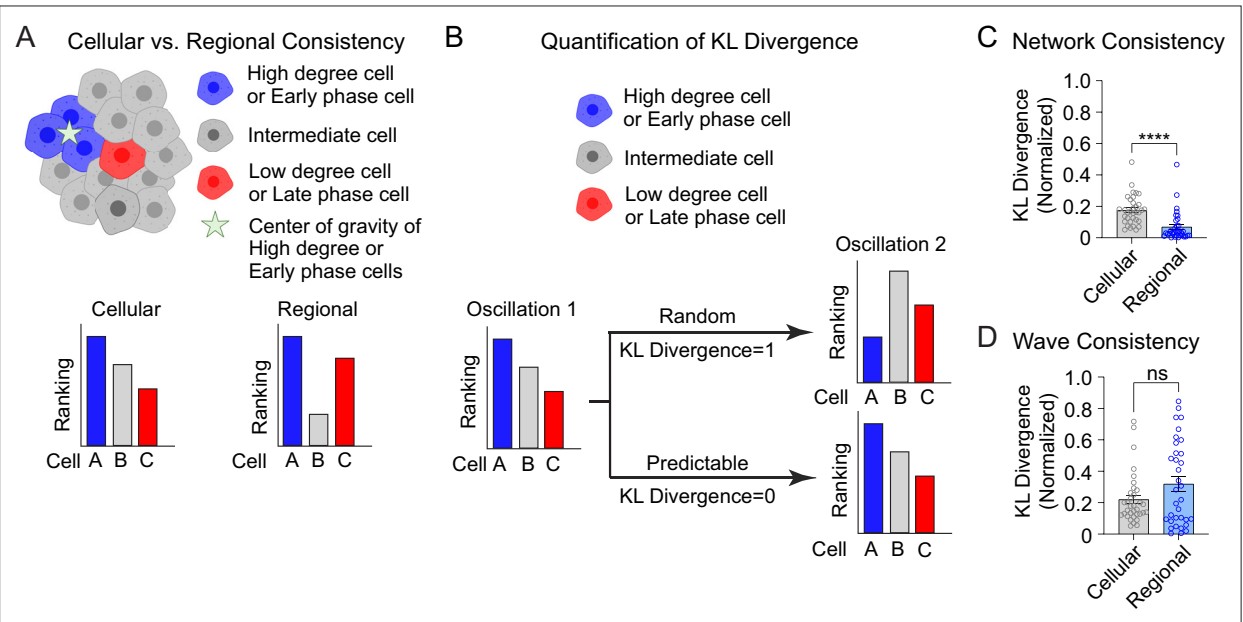

**Figure 4.** Cellular and regional consistency of the β-cell network and Ca²⁺ wave quantified by Kullback–Leibler (KL) divergence. (A) Schematic showing cellular and regional consistency analyses. (B) Schematic depicting the use of KL divergence to determine consistency between consecutive oscillations. Every β-cell in the islet is ranked, with near-zero KL divergence values indicating high consistency between oscillations and near-unity KL divergence indicating randomness. Comparison of cellular versus regional consistency of the network (C) and wave (D) by KL divergence. Data are displayed as mean ± SEM. ****p < 0.0001 by normality test followed by Paired Student's *t*-test or Wilcoxon Signed-Rank Test.

The online version of this article includes the following source data for figure 4:

**Source data 1.** Source for network and wave consistency analysis.

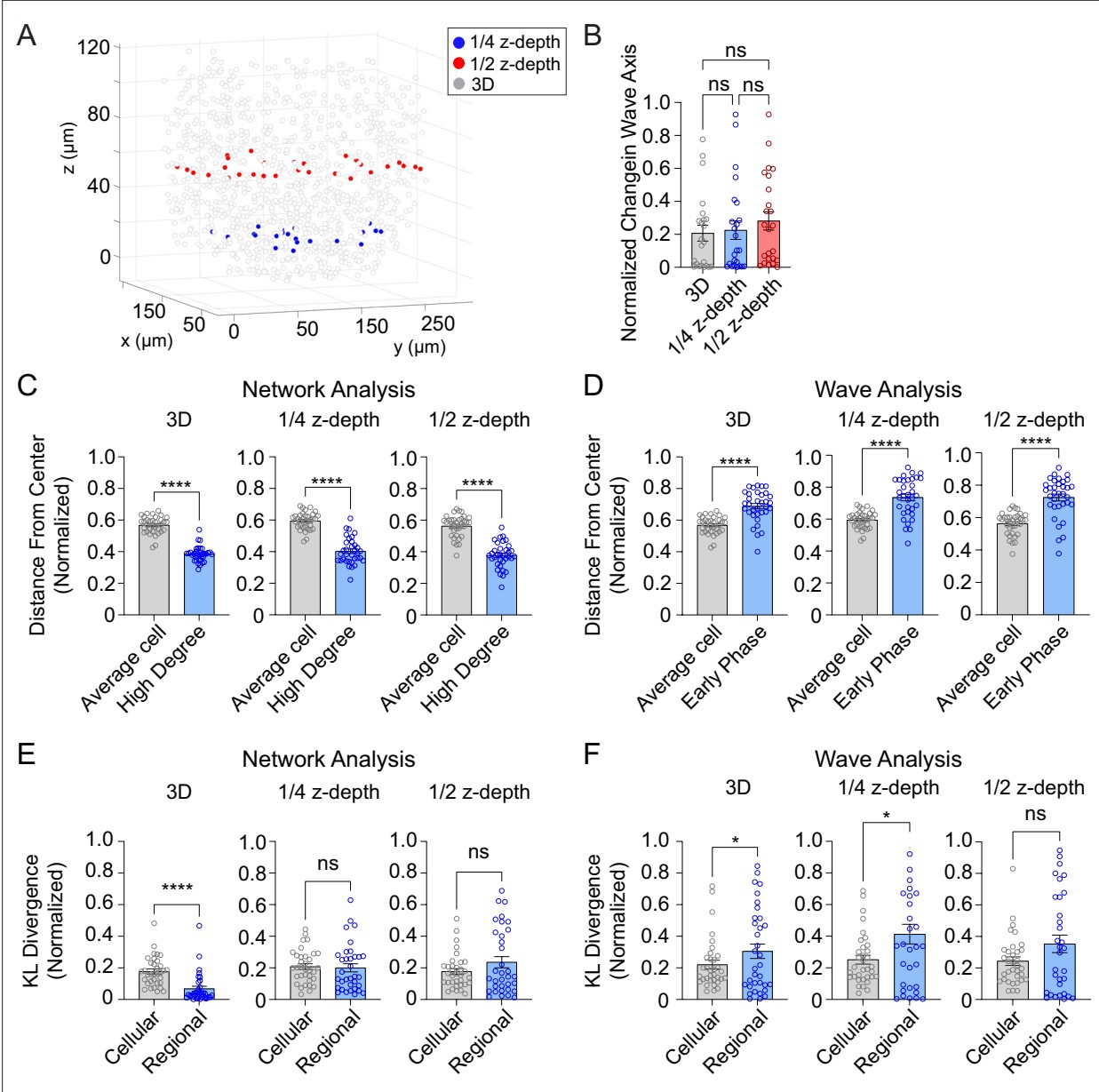

**Figure 5.** Three-dimensional (3D) analysis is more robust than two-dimensional (2D) analysis. (**A**) Example islet showing the locations of the ¼-depth (red) and ½-depth (blue) 2D planes used for analysis. (**B**) Comparison of wave axis change from 2D and 3D analyses. (**C**) Comparison of distance from center for average and high degree cells based on either 3D (left panel) or 2D planes (middle and right panels). (**D**) Comparison of distance from center for average and early phase cells based on either 3D (left panel) or 2D planes (middle and right panels). (**E, F**) Comparison of cellular and regional consistency of the network (**E**) and Ca²⁺ wave (**F**) based on either 3D (left panel) or 2D planes (middle and right panels). Data are displayed as mean ± SEM. *p < 0.05, ****p < 0.0001 by normality test followed by parametric or non-parametric one-way ANOVA (**B**) or Student's *t*-test or Wilcoxon Signed-Rank Test (**C–F**).

The online version of this article includes the following source data for figure 5:

**Source data 1.** Source data for 3D and 2D analysis comparison.

wave was consistent cellularly, this analysis may imply that the wave is established by cellular properties, whereas the network is emergent.

## The consistency of 2D analyses of the network and wave is much lower than 3D analyses

To investigate whether 2D analysis, as performed in all prior studies (*Johnston et al., 2016*; *Westacott et al., 2017*) provides a similar level of robustness as the current 3D analysis, we performed network and wave analyses on a single plane at either ¼- or ½-depth of the z-stack (*Figure 5A*). Both the 2D and 3D analyses showed that the wave axis changes over time (*Figure 5B*). The analyses also agreed that the high degree cells are located at the center of the islet (*Figure 5C*) and that the early phase cells are located at the edge of the islet (*Figure 5D*). However, when we looked at the regional and cellular consistency of the β-cell network, the 2D analysis at both ¼- and ½-depth of the z-stack showed no difference for regional and cellular consistency (*Figure 5E*). This result contradicts with the 3D analysis which showed the regional consistency of the β-cell network is significantly more stable than cellular consistency. When analyzing the wave, both 3D and 2D analyses at ¼-depth showed that the cellular consistency is more stable than regional consistency, while the results from a plane at ½-depth showed no difference (*Figure 5F*). These findings indicate that 2D imaging at different planes of the islet can sometimes skew the results of the heterogeneity analysis.

## The origin of Ca²⁺ waves in 3D space is determined by the activity of glucokinase, while the β-cell network is patterned independently of metabolic input

Glycolysis exerts strong control over the timing of β-cell Ca²⁺ oscillations (*Merrins et al., 2022*; *Tornheim, 1997*). Glucokinase, as the 'glucose sensor' for the β-cell, controls the input of glucose carbons into glycolysis (*Merrins et al., 2022*; *Matschinsky and Ellerman, 1968*), and the downstream action of pyruvate kinase controls membrane depolarization by closing $K_{ATP}$ channels (*Merrins et al., 2022*; *Lewandowski et al., 2020*; *Foster et al., 2022*; *Ho et al., 2023*; *Quesada et al., 1999*; *Figure 6A*). We applied glucokinase activator (GKa, 50 nM RO-28-1675) and pyruvate kinase activator (PKa, 10 µM TEPP-46) (*Lewandowski et al., 2020*; *Foster et al., 2022*) to determine the effects of these enzymes on β-cell subpopulations during glucose-stimulated oscillations.

Since biochemically distinct processes occur during the silent phase (i.e., the electrically silent period when $K_{ATP}$ channels close and Ca²⁺ remains low) and the active phase (i.e., the electrically active period Ca²⁺ is elevated and secretion occurs) (*Merrins et al., 2022*), we quantified the duration of each phase along with the oscillation period and duty cycle (the ratio of active phase to full cycle) (*Figure 6B*). In the presence of vehicle control (0.1% DMSO), the Ca²⁺ duty cycle remained stable, although an outlier in the control group resulted in a small decrease in the silent phase duration (*Figure 6C*). In 3 of 15 islets, GKa induced a Ca²⁺ plateau (duty cycle = 1.0); out of necessity these islets were removed from the oscillation analysis. In the majority of islets, GKa increased the oscillation period and duty cycle (*Figure 6D*). The duty cycle increase was driven by an increase in the active phase duration, with no impact on the silent phase, the time when $K_{ATP}$ channels close (*Figure 6D*). In contrast with activation of glucokinase, PKa increased the oscillation frequency by reducing the silent and the active phase duration in equal proportions (*Figure 6E*). The absence of any PKa effect on the duty cycle is expected since fuel input is controlled by glucokinase, whereas silent phase shortening is expected based on the ability of PKa to reduce the time required to close $K_{ATP}$ channels and depolarize the plasma membrane (*Lewandowski et al., 2020*; *Foster et al., 2022*). Thus, a single-cell 3D analysis of β-cell Ca²⁺ oscillation upon GKa and PKa stimulation provides similar conclusions to prior 2D studies of intact islets.

Previous 2D studies have found metabolic differences along the Ca²⁺ wave, as measured by NAD(P)H fluorescence (*Westacott et al., 2017*). We measured the 3D position of early or late phase cells in response to glucokinase or pyruvate kinase activation. A positional analysis showed that GKa strongly reinforced the islet region corresponding to early and late phase cells, while vehicle and PKa had no discernable effect (*Figure 7A*). The KL divergence for wave propagation was correspondingly reduced by GKa (*Figure 7B*), indicating increased consistency, and the wave axis was significantly stabilized by GKa (*Figure 7C*). Again, PKa had no discernable effect on the KL divergence for wave propagation or

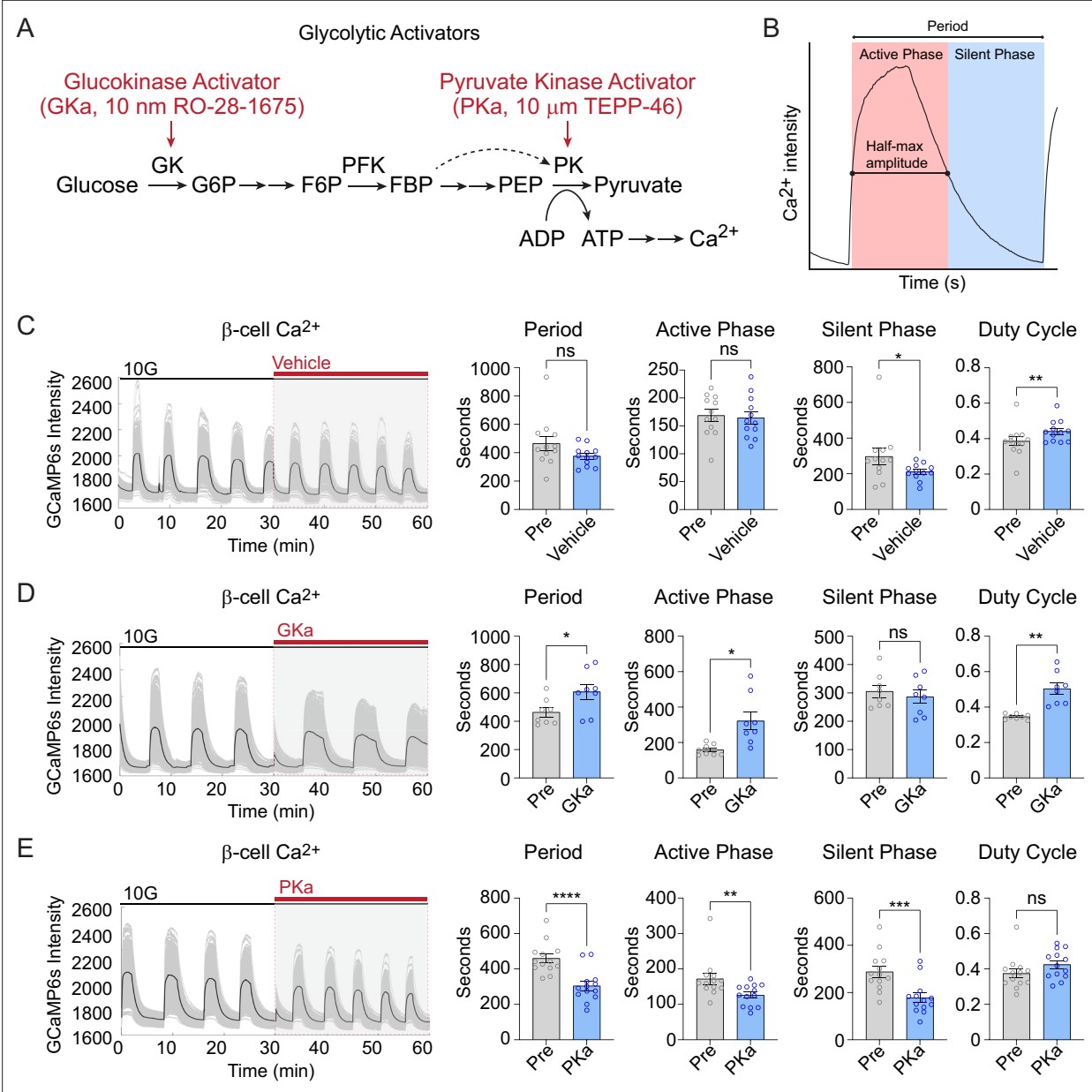

**Figure 6.** Effect of glycolytic activators on β-cell oscillations. (**A**) Schematic of glycolysis showing the targets of glucokinase activator (GKa) and pyruvate kinase activator (PKa). (**B**) Illustration indicating the oscillation period, active phase duration, silent phase duration, and duty cycle (active phase/period) calculated at half-maximal Ca²⁺. Sample traces and comparison of period, active phase duration, silent phase duration, and duty cycle before and after vehicle (0.1% DMSO) (n = 11,284 cells, 13 islets, 7 mice) (**C**), GKa (50 nM RO-28-1675) (n = 6871 cells, 8 islets, 7 mice) (**D**), and PKa (n = 10,700 cells, 13 islets, 7 mice) (10 μM TEPP-46) (**E**). Data are displayed as mean ± SEM. *p < 0.05, **p < 0.01, ***p < 0.001, ****p < 0.0001 normality test followed by Paired Student's *t*-test or Wilcoxon Signed-Rank Test.

The online version of this article includes the following source data for figure 6:

**Source data 1.** Source data for period, active phase, silent phase and duty cycle analysis.

wave axis stability, a likely indication that the Ca²⁺ wave origin is primarily, if not exclusively, controlled by glucokinase patterning.

As a second approach, we examined the percentage of early or late phase cells maintained following activation of glucokinase or pyruvate kinase. Early phase cells were maintained to a greater degree upon GKa application, indicating greater consistency, but again showed no change upon vehicle or PKa application (*Figure 7D*). Late phase cells showed no difference in their maintenance upon any of the treatments (*Figure 7E*), suggesting that the earliest phase cells drive the consistency

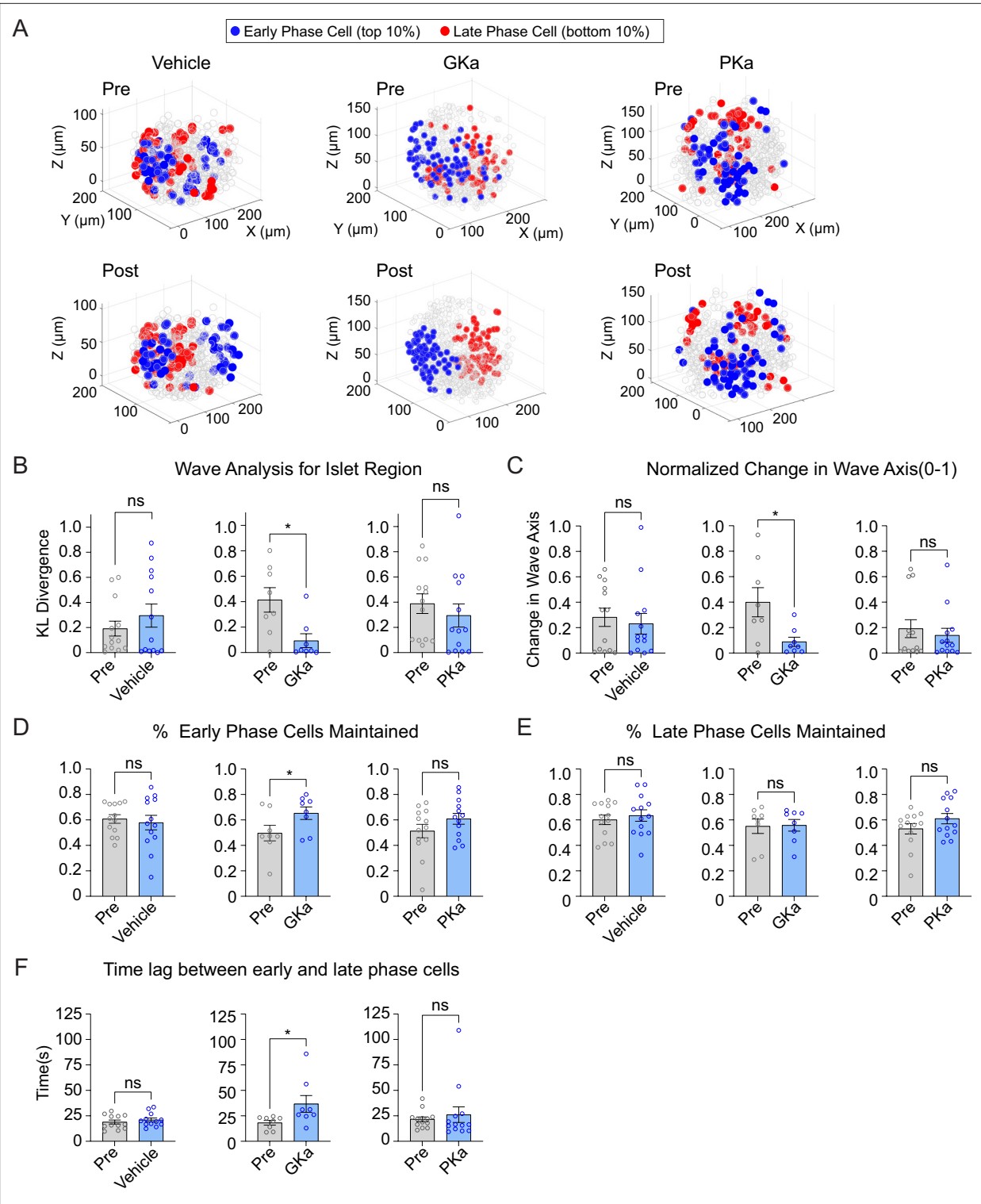

**Figure 7.** Glucokinase activity determines the origin of Ca²⁺ waves in three-dimensional (3D) space. (**A**) Illustrations showing the location change of early phase cells (blue) and late phase cells (red) before and after vehicle (left panel), GKa (middle panel), and PKa (right panel). Effect of vehicle, GKa, and PKa on regional consistency of the Ca²⁺ wave (**B**), wave axis change (**C**), early phase cell retention (**D**), late phase cell retention (**E**), and the time lag between early and late phase cells (**F**). Data are displayed as mean ± SEM. *p < 0.05 by Student's *t*-test.

The online version of this article includes the following source data and figure supplement(s) for figure 7:

**Source data 1.** Source data for islet region, wave axis, % early/late phase cells maintained and time lag.

*Figure 7 continued on next page*

*Figure 7 continued*

**Figure supplement 1.** Effect of glycolytic activators on the β-cell network.

**Figure supplement 1—source data 1.** Source data for islet region, % high/low degree cells maintained.

**Figure supplement 2.** Effect of glycolytic activators on the β-cells that depolarize first.

**Figure supplement 2—source data 1.** Source data for % early depolarizer maintained, regional consistency and wave axis change.

of the wave propagation. The time lag between early and late phase cells was increased upon GKa application (*Figure 7F*), showing that GK activation can enlarge the differences between early and late phase cells.

While metabolic differences have been suggested to underlie functional heterogeneity in the β-cell network (*Johnston et al., 2016*; *Briggs et al., 2023*), we observed no changes in the consistency of the islet network upon either GKa or PKa application (*Figure 7—figure supplement 1A*). Similarly, the consistency of high or low degree cells also did not change upon either GKa or PKa application (*Figure 7—figure supplement 1B, C*). Collectively, these findings implicate glucokinase as the key determinant of the $Ca^{2+}$ wave in 3D space, whereas metabolic perturbations have little influence on the islet network.

We defined early phase cells as those that depolarized and repolarized first. We also assessed whether the results were consistent for cells that only depolarized first (while ignoring repolarization). Similar to early phase cells, GKa increased the retention of β-cells that depolarized first (*Figure 7—figure supplement 2A*) and their regional consistency (*Figure 7—figure supplement 2B*). However, GKa did not influence the wave axis change (*Figure 7—figure supplement 2C*), indicating that the cells that depolarize first are a more unstable population than those that depolarize and repolarize first.

## Discussion

In this study, we used light-sheet microscopy of single mouse islets to provide a 3D analysis of the β-cell subpopulations that initiate $Ca^{2+}$ oscillations and coordinate the islet network. While this ex vivo approach might not precisely mimic the in vivo situation, our analyses show that 3D imaging is a more robust approach than 2D imaging, which does not accurately reflect heterogeneity and subpopulation consistency over the entire islet. Reinforcing the concept of distinct β-cell subpopulations, the most highly synchronized cells are located at the center of the islet, while those β-cells that control the initiation and termination of $Ca^{2+}$ waves (leaders) were located on the islet periphery. We further observed that different regions of the islet initiate the $Ca^{2+}$ wave over time, challenging the view that leader cells are a fixed pacemaker population of cells defined by their biochemistry. As discussed below, technical advances in image capture and analysis provide several new insights into the features of β-cell subpopulation in 3D and illustrate how glycolytic enzymes influence the system.

β-Cell $Ca^{2+}$ imaging is an indispensable approach for understanding pulsatility. When studied by light-sheet imaging, islets exhibited similar ~3- to 5-min oscillations as in vivo two-photon imaging of β-cell $Ca^{2+}$ oscillations in live mice (*Adams et al., 2021*), as well as the high-speed confocal imaging used in prior ex vivo studies (*Peng et al., 2024*). Light-sheet imaging overcomes the speed and depth limitations, respectively, that prevent these approaches from single-cell analysis of the entire islet. Relative to spinning disk confocal, penetration depth increased >twofold with the light-sheet microscope (from 50–60 to 130–150 μm), allowing small- and medium-sized islets to be imaged in toto. Abandoning confocal pinholes improved light collection, and therefore acquisition speed, ~threefold; this is an underestimate given the 0.4 $e^-$ read noise cameras on the spinning disk microscope versus 1.6 $e^-$ read noise cameras on the light-sheet microscope. The ~1.1-μm axial resolution of the light-sheet, while lower than spinning disk confocal, was easily sufficient for Nyquist sampling of 5–6 μm nuclei used to identify each β-cell in 3D space (β-cells themselves are 12–18 μm). Together these features allowed sampling the islet at >2 Hz, although future studies could be improved by employing a higher NA objective and a camera with lower read noise and higher quantum efficiency.

Phase and functional network analyses were used to understand the behavior of β-cell subpopulations and how they communicate. Importantly, in past heterogeneity studies, phase and network calculations were assessed over the entire time course (*Johnston et al., 2016*; *Dwulet et al., 2021*;

*Hraha et al., 2014*; *Stožer et al., 2022*; *Stožer et al., 2013b*). Here, we assessed over individual oscillations to compare subpopulation stability over time. One population of cells are those which we termed 'early phase cells' and lead the propagating $Ca^{2+}$ wave. These have also been referred to as 'leader cells' or 'pacemaker cells' and regulate the oscillatory dynamics (*Salem et al., 2019*; *Peng et al., 2024*; *Peng et al., 2024*). To our surprise, the early phase cells (i.e., the leader cells) were not consistent over time. The late phase cells, located on the opposite end of the islet, showed a similar shift, with over half of the islets showing changes in the wave axis. Consequently, laser ablation of these early or late phase cells would be predicted to have little impact on islet function, as suggested previously by electrophysiological studies in which surface β-cells have been voltage-clamped with no impact on β-cell oscillations (*Satin et al., 2020*), or computational studies in which removal of simulated β-cells had little impact on resulting oscillations (*Dwulet et al., 2021*; *Korošak et al., 2021*).

Studies have sought to define whether β-cell intrinsic or extrinsic factors determine the oscillations (*Johnston et al., 2016*; *Satin et al., 2020*; *Briggs et al., 2023*; *Šterk et al., 2023*). Because of the shift in $Ca^{2+}$ wave axis between consecutive oscillations, we conclude that β-cell depolarization is dominated by stochastic properties rather than a pre-determined genetic or metabolic profile. Previous experimental and modeling studies have suggested that the $Ca^{2+}$ wave origin corresponds to the glucokinase activity gradient (*Dwulet et al., 2021*; *Hraha et al., 2014*). Consistent with this prediction, pharmacologic activation of glucokinase reinforced the islet region of early phase cells and reduced the wave axis change. Pyruvate kinase activation, despite increasing oscillation frequency, had no effect on leader cells, indicating the wave origin is patterned by fuel input. Importantly, there is no evidence that the glucokinase gradient is the result of intentional spatial organization. Rather, in computational studies the glucokinase gradient emerges stochastically due to randomly placed high and low glucokinase-expressing cells, with multiple competing glucokinase gradients determining the degree of wave axis rotation. Our findings suggest that when glucokinase is activated, the strongest gradient is amplified, which is why the $Ca^{2+}$ wave axis is reinforced. Another compelling hypothesis for stochastic behavior, which is not mutually exclusive, is the heterogenous nutrient response of neighboring α-cells influences the excitability of neighboring β-cells via GPCRs (*Capozzi et al., 2019*; *El et al., 2021*; *Kang et al., 2008*). The preponderance of α-cells on the periphery of mouse islets, which influence β-cell oscillation frequency (*Ren et al., 2022*), would be expected to disrupt β-cell synchronization on the periphery and stabilize it in the islet center – which is precisely the pattern of network activity we observed. In addition to α-cells, vasculature may also impact islet $Ca^{2+}$ responses (*Jevon et al., 2022*), and may induce additional heterogeneity in vivo.

Functional network studies of the islet revealed a heterogeneity in β-cell functional connections (*Stožer et al., 2013b*). A small subpopulation of β-cells, termed 'hub' cells, was found to have the highest synchronization to other cells (*Johnston et al., 2016*). Optogenetic silencing of hub cells was found to disrupt network activity within that plane, however it should be noted that hub cells are defined as the most highly coordinated cells within a randomly selected plane of the islet. Debates exist over whether the hub cells can maintain electrical control over the whole islet (*Satin et al., 2020*; *Satin and Rorsman, 2020*). Because our study investigated the 3D β-cell functional network over individual oscillations, our top 10% of highly coordinated cells are not the exact same population as hub cells defined in *Johnston et al., 2016*, however the subpopulations likely overlap. In contrast with leader cells, we found that the highly synchronized hub cells are both spatially and temporally stable. However, in conflict with the description of hub cells as intermingled with other cells throughout the islet (*Johnston et al., 2016*), the location of such cells in 3D space is close to the center. This observation could be explained by the peripheral location of α-cells as discussed above for the $Ca^{2+}$ wave behavior.

Previous studies indicated that the intrinsic metabolic activity and thus oscillation profile may play a larger role in driving high synchronization than the strength of gap junction coupling (*Briggs et al., 2023*; *Šterk et al., 2023*). This included experimental 2D measurements but also computational 3D measurements. Nevertheless, we demonstrated here that perturbing glucokinase or pyruvate kinase had little effect on the consistency of the high or low degree cells within the 3D network. We further observed that the β-cell network was more regionally consistent than cellularly consistent, indicating a tendency for nearby cells within the islet center to 'take over' as high degree cells. The mechanisms underlying this are unclear. One explanation may be that paracrine communication within the islet determines which region of cells will show high or low degree (*Ren et al., 2022*). For example,

more peripheral cells that are in contact with nearby δ-cells may show some suppression in their $Ca^{2+}$ dynamics (*Dickerson et al., 2022*), and thus reduced synchronization. Alternatively, more peripheral cells may show increased stochastic behavior that reduces their relative synchronization. Modulating α/δ-cell inputs to the β-cell in combination with 3D islet imaging will be important to test this in the future. Our study emphasizes that 3D studies are critical to fully assess the consistency and spatial organization of the β-cell network.

## Methods
### Mice
*Ins1-Cre* mice (*Thorens et al., 2015*) (Jax 026801) were crossed with *GCaMP6s* mice (Jax 028866), a Cre-dependent $Ca^{2+}$ indicator strain, and *H2B-mCherry* mice, a Cre-dependent nuclear indicator strain (*Blum et al., 2014*). The resulting *Ins1-Cre:ROSA26$^{GCaMP6s/H2B-mCherry}$* mice were genotyped by Transnetyx. Mice were sacrificed by $CO_2$ asphyxiation followed by cervical dislocation at 12–15 weeks of age, and islets were isolated and cultured as detailed in *Ho et al., 2023*. All procedures involving animals were approved by the Institutional Animal Care and Use Committees of the William S. Middleton Memorial Veterans Hospital and followed the NIH Guide for the Care and Use of Laboratory Animals.

### Light-sheet microscope
The stage of a Nikon Ti inverted epifluorescence microscope was replaced with a Mizar TILT M-21N lateral-interference tilted excitation light-sheet generator (*Fadero et al., 2018*) equipped with an ASI MS-2000 piezo z-stage and Okolab stagetop incubator. The sample chamber consisted of an Ibidi 4-well No. 1.5 glass bottom chamber slide with optically clear sides. Excitation from a Vortran Stradus VeraLase 4-channel (445/488/561/637) single mode fiber-coupled laser and CDRH control box (AVR Optics) was passed through the TILT cylindrical lens to generate a light-sheet with a beam waist of 4.3 µm directly over the objective's field of view. Similar to a widefield microscope, the axial (z) resolution of the light-sheet microscope is dictated by the NA of the objective (~1.1 µm for our Nikon CFI Apo LWD Lambda S 40XC water immersion objective with an NA of 1.15). Fluorescence emission was passed through an optical beamsplitter (OptoSplit III, 89 North) and collected by an ORCA-Flash4.0 v3 digital CMOS camera (Hamamatsu C13440-20CU) equipped with a PoCL camera link cable. To achieve high-speed triggered acquisition, the laser and piezo z-stage were triggered directly by the camera, which received a single packet of instructions from the NIS-Elements JOBS module via a PCI express NiDAQ card (PCIe-6323, National Instruments). Electronic components (DAQ card, camera, lasers, stage) were linked by a Nikon 'standard cable' via a Nikon BNC breakout box; cable assembly is diagrammed in *Figure 1—figure supplement 1*. Images were streamed to a Dell computer equipped with an Intel Xeon Silver 4214R CPU, 256 GB RAM, XG5 NVMe SSD, and NVIDIA Quadro Pro 8 GB graphics card and Bitplane Imaris Software (Andor).

### Imaging of β-cell $Ca^{2+}$ and nuclei
Reagents were obtained from Sigma-Aldrich unless indicated otherwise. Islets isolated from *Ins1-Cre:ROSA26$^{GCaMP6s/H2B-mCherry}$* mice were incubated overnight and loaded into an Ibidi µ-slide 4-well No. 1.5 glass bottom chamber slide and maintained by an Okolab stagetop incubator at 37°C. The bath solution contained, in mM: 135 NaCl, 4.8 KCl, 2.5 $CaCl_2$, 1.2 $MgCl_2$, 20 HEPES, 10 glucose, 0.18 glutamine, 0.15 leucine, 0.06 arginine, 0.6 alanine, pH 7.35. Glucokinase activator (50 nM RO-28-1675, Axon), pyruvate kinase activator (10 µM TEPP-46, Calbiochem), and vehicle control (0.1% DMSO) were added as indicated. GCaMP6s (488 nm, 5% power, 50 mW Vortran Stradus Versalase) and H2B-mCherry (561 nm, 20% power, 50 mW) were simultaneously excited and emission was simultaneously collected on a single camera chip using an optical beamsplitter (Optosplit III, 89 North) containing a dichroic mirror (ZT568rdc, Chroma) and emission filters for GCaMP6s (ET525/40, Chroma) and mCherry (ET650/60, Chroma). The exposure time was set to 15 ms in NIS-Elements JOBS, which includes ~10 ms camera integration time and ~5 ms stage dwell time. This was sufficiently fast to image intact islets an axial (z) depth of 132 µm at 2.02 Hz (33 z-steps every 4 µm). Raw NIS-Elements ND2 files were imported into Bitplane Imaris analysis software. The location of each cell was marked using H2B-mCherry nuclear signal and a sphere mask was created based on average β-cell nuclear

diameter. Nuclear ROIs were mathematically expanded to 9.3 µm to avoid overlapping cells, which was determined by point-scanning confocal imaging (**Briggs et al., 2024**). Masks were propagated to all the time points and mean Ca²⁺ levels were used to generate single-cell traces that were exported from Imaris to Microsoft Excel. Quantitative analyses of the β-cell network and Ca²⁺ wave were performed in MATLAB as described below.

### Identification of β-cell Ca²⁺ oscillations

To compare β-cell subpopulations over multiple oscillations, we developed a semi-automated oscillation identifier to ensure that the results did not depend on manual identification of oscillation start and end times. First, the approximate time corresponding to the peak of each oscillation was manually identified based on the average islet signal. The time course around each oscillation peak was automatically extracted as seconds before the islet begins depolarization $x/2$ s after the islet completes repolarization, where $x = ¼$ oscillation duty cycle. Depolarization and repolarization were then automatically identified using the derivatives of the Ca²⁺ time course and MATLAB's findpeaks function. All oscillation time courses were manually confirmed. For studies of glycolysis, we ensured that all pre- and post-glycolytic activator treatments had the same number of oscillations. All islets analyzed exhibited slow Ca²⁺ oscillations (period = 6.77 ± 0.36 min).

### Network analysis

Network analysis was conducted as described in **Briggs et al., 2023**, with the caveat that the functional network was recalculated for each oscillation. The correlation threshold was calculated such that the average degree was 7 when averaged over all oscillations (**Šterk et al., 2024**).

### Wave analysis

Lagged cross correlation between the normalized Ca²⁺ dynamics of each cell and the islet mean was calculated for each oscillation. Each cell was assigned a cell phase, defined as the time lag that maximized the cross correlation.

### Wave axis

Wave axis was defined as the primary axis between the early (top 10%) and late (bottom 10%) of cells in the Ca²⁺ wave. The primary axis was identified using principal component analysis. To ensure the axis was not confounded by spuriously located cells, cells were not included in the analysis if they were greater than 50 µm from the center of gravity (calculated using Euclidean distance) of their respective group (early or late phase). Variability of the wave axis over oscillations was defined as the squared Euclidean distance between each wave axis. To compare across islets of differences sizes, wave axis variability was normalized by the maximum variability possible for each islet. This maximum variability was identified by repeating the wave axis calculation 50,000 times for randomly selected early and late phase cells.

### KL divergence

To calculate consistency over oscillations of the entire islet, network and wave analyses were conducted and cells were ranked for each oscillation ($i$) based on (1) their degree or phase (for cellular consistency) and (2) their Euclidean distance to the center of gravity (for regional consistency) of the high degree or early phase cells. The probability density functions ($P_i$) of these rankings were calculated using the MATLAB normpdf function. The KL divergence (**Kullback and Leibler, 1951**) ($D_{KL}$) between cell ranks for each oscillation was calculated, where ($n$) is the index of each cell, and ($i,j$) are indices for each oscillation.

$$D_{KL}\left(P_i \parallel P_j\right) = \sum_n P_i \log \frac{P_i}{P_j}, \tag{1}$$

To compare across islets of differences sizes, we normalized the KL divergence by the maximum possible KL divergence for each islet identified by shuffling the cell ranks and calculating KL divergence 100 times.

## Comparison of 2D and 3D analyses

Quarter- and half-depth 2D planes were selected from each 3D islet. Cells were included in the 2D plane if their location on the z-axis was within 3 μm of the plane.

## Statistical analysis

Stastistical analysis was conducted using GraphPad PRISM 9.0 software. Significance was tested by first testing normality using Anderson–Darling and Kolmogorov–Smirnov normality tests and then using paired Wilcox tests, Student's two-tailed t-tests or ANOVA as indicated. $p < 0.05$ was considered significant and errors signify ± SEM.

## Acknowledgements

We thank Barak Blum at the University of Wisconsin-Madison for providing $ROSA26^{H2B-mCherry}$ mice and the University of Wisconsin Optical Imaging Core for use of the spinning disk confocal. The Merrins laboratory gratefully acknowledges support from the NIH/NIDDK (R01DK113103 and R01DK127637 to MJM, and R01DK106412 to RKPB) and the United States Department of Veterans Affairs Biomedical Laboratory Research and Development Service (I01BX005113 to MJM). The Benninger laboratory gratefully acknowledges support from the NIH/NIDDK (R01DK106412, R01DK102950, R01DK140904 to RKPB) and the University of Colorado Diabetes Research center (P30 DK116073). Jennifer K Briggs acknowledges support from NSF GRFP (DGE-1938058_Briggs).

## Additional information

### Funding

| Funder | Grant reference number | Author |
|---|---|---|
| National Institute of Diabetes and Digestive and Kidney Diseases | R01DK113103 | Matthew J Merrins |
| National Institute of Diabetes and Digestive and Kidney Diseases | R01DK127637 | Matthew J Merrins |
| National Institute of Diabetes and Digestive and Kidney Diseases | R01DK106412 | Richard KP Benninger |
| Biomedical Laboratory Research and Development, VA Office of Research and Development | I01BX005113 | Matthew J Merrins |
| National Science Foundation | DGE-1938058_Briggs | Jennifer K Briggs |
| National Institute of Diabetes and Digestive and Kidney Diseases | R01DK102950 | Richard KP Benninger |
| National Institute of Diabetes and Digestive and Kidney Diseases | R01DK140904 | Richard KP Benninger |
| University of Colorado | P30 DK116073 | Richard KP Benninger |

The funders had no role in study design, data collection and interpretation, or the decision to submit the work for publication.

### Author contributions

Erli Jin, Conceptualization, Data curation, Formal analysis, Investigation, Methodology, Writing – original draft, Writing – review and editing; Jennifer K Briggs, Conceptualization, Software, Formal

analysis, Investigation, Visualization, Methodology, Writing – review and editing; Richard KP Benninger, Conceptualization, Resources, Supervision, Funding acquisition, Project administration, Writing – review and editing; Matthew J Merrins, Conceptualization, Resources, Supervision, Funding acquisition, Writing – original draft, Project administration, Writing – review and editing

### Author ORCIDs
Erli Jin ⬥ https://orcid.org/0009-0002-9410-9738
Jennifer K Briggs ⬥ https://orcid.org/0000-0002-8737-2215
Richard KP Benninger ⬥ https://orcid.org/0000-0002-5063-6096
Matthew J Merrins ⬥ https://orcid.org/0000-0003-1599-9227

### Ethics
All procedures involving animals were approved by the Institutional Animal Care and Use Committees of the William S. Middleton Memorial Veterans Hospital and followed the NIH Guide for the Care and Use of Laboratory Animals.

Reviewer #1 (Public review): https://doi.org/10.7554/eLife.103068.3.sa1
Reviewer #2 (Public review): https://doi.org/10.7554/eLife.103068.3.sa2
Reviewer #3 (Public review): https://doi.org/10.7554/eLife.103068.3.sa3
Author response https://doi.org/10.7554/eLife.103068.3.sa4

---

## Additional files

### Supplementary files
MDAR checklist

Source data 1. Raw data for single cell traces (10G 1/3).

Source data 2. Raw data for single cell traces (10G 2/3).

Source data 3. Raw data for single cell traces (10G 3/3).

Source data 4. Raw data for single cell traces (10G-10GGKa 1/2).

Source data 5. Raw data for single cell traces (10G-10GGKa 2/2).

Source data 6. Raw data for single cell traces (10G-10GPKa 1/3).

Source data 7. Raw data for single cell traces (10G-10GPKa 2/3).

Source data 8. Raw data for single cell traces (10G-10GPKa 3/3).

### Data availability
All data generated or analyzed during this study are included in the manuscript and supporting files. All code is publicly available at GitHub (copy archived at *Briggs, 2024*).

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
