## [Editor Report · eLife Assessment]

This study provides **compelling** evidence for functional subpopulations of β-cells responsible for Ca2+ signal initiation and maintenance using novel three-dimensional light sheet microscopy imaging and analysis of pancreatic islets. The findings are **important** as they help decode mechanistic underpinnings of islet calcium oscillations and the resulting pulsatile insulin secretion. The work will be of general interest to cell biologists and particular interest to islet biologists.

---

## [Referee Report · Reviewer #1 (Public review)]

Summary:

Jin, Briggs and colleagues use light sheet imaging to reconstruct the islet three-dimensional Ca2+ network. The authors find that early/late responding (leader) cells are dynamic over time, and located at the islet periphery. By contrast, highly connected or hub cells are stable, and located toward the islet center. Suggesting that the two subpopulations are differentially regulated by fuel input, glucokinase activation only influences leader cell phenotype, whereas hubs remain stable.

Strengths:

The studies are novel in providing the first three-dimensional snapshot of the beta cell functional network, as well as determining the localization of some of the different subpopulations identified to date. The studies also provide some consensus as to the origin, stability and role of such subpopulations in islet function.

Weaknesses:

Experiments with metabolic enzyme activators do not take into account the influence of cell viability on the observed Ca2+ network data. Limitations of the imaging approach used need to be recognised and evaluated/discussed.

Comments on revisions:

The authors have addressed the majority of the points raised.

---

## [Referee Report · Reviewer #2 (Public review)]

The manuscript by Erli Jin and Jennifer Briggs et al. utilizes light sheet microscopy to image islet beta cell calcium oscillations in 3D and determine where beta cell populations are located that begin and coordinate glucose-stimulated calcium oscillations. The light sheet technique allowed clear 3D mapping of beta cell calcium responses to glucose, glucokinase activation, and pyruvate kinase activation. The manuscript finds that synchronized beta-cells are found at the islet center, that leader beta cells showing the first calcium responses are located on the islet periphery, that glucokinase activation helped maintain beta cells that lead calcium responses, and that pyruvate kinase activation primarily increases islet calcium oscillation frequency. The study is well-designed, contains a significant amount of high quality data, and the conclusions are largely supported by the results.

Comments on revisions:

The manuscript by Erli Jin et al. has been improved with the revisions, which have addressed my previous concerns. The manuscript significantly improves the mechanistic underpinnings of islet calcium oscillations and resulting pulsatile insulin secretion.

---

## [Referee Report · Reviewer #3 (Public review)]

Summary:

Jin, Briggs et al. made use of light-sheet 3D imaging and data analysis to assess the collective network activity in isolated mouse islets. The major advantage of using whole islet imaging, despite compromising on a speed of acquisition, is that it provides a complete description of the network, while 2D networks are only an approximation of the islet network. In static-incubation conditions, excluding the effects of perfusion, they assessed two subpopulations of beta cells and their spatial consistency and metabolic dependence.

Strengths:

The authors confirmed that coordinated Ca2+ oscillations are important for glycemic control. In addition, they definitively disproved the role of individual privileged cells, which were suggested to lead or coordinate Ca²⁺ oscillations. They provided evidence for differential regional stability, confirming the previously described stochastic nature of the beta cells that act as strongly connected hubs as well as beta cells in initiating regions (doi.org/10.1103/PhysRevLett.127.168101). This has not been a surprise to the reviewer.

The fact that islet cores contain beta cells that are more active and more coordinated has also been readily observed in high-frequency 2D recordings (e.g. DOI: 10.2337/db22-0952), suggesting that the high-speed capture of fast activity can partially compensate for incomplete topological information.

They also found an increased metabolic sensitivity of mantle regions of an islet with subpopulation of beta cells with a high probability of leading the islet activity and which can be entrained by fuel input. They discuss a potential role of alpha/delta cell interaction, however relative lack of beta cells in the islet border region could also be a factor contributing to less connectivity and higher excitability.

The Methods section contains a useful series of direct instructions on how to approach fast 3D imaging with currently available hardware and software.

The Discussion is clear and includes most of the issues regarding the interpretation of the presented results.

Taken together it is a strong technical paper to demonstrate the stochasticity regarding the functions subpopulations of beta cells in the islets may have and how less well-resolved approaches (both missing spatial resolution as well as missing temporal resolution) led us to jump to unjustified conclusions regarding the fixed roles of individual beta cells within an islet.

Weaknesses:

There are a few relevant issues that need to be addressed.

(1) The study is not internally consistent regarding the Results section. In the text the authors discuss changes in membrane potential (not been measured in this study), while in the figures they exclusively describe Ca2+ oscillations (which were measured). Examples are on lines 149, 150, 153, 154, 263... It is recommended that the silent and active phase in the Results section describe processes actually measured in this study as shown 6A.

(2) There are in fact no radially oriented networks in the core of an islet (l. 130, Fig. 4) apart from the fact that every hub has somewhat radially oriented edges. For radiality to have some general meaning, the normalized distance from the geometric center would need to be lower than 0.4. The networks are centrally located, which does not change the major conclusions of the study.

(3) The study would profit from acknowledging that Ca2+ influx is not a sole mechanism to drive insulin secretion and that KATP channels are not the sole target sensitive to changes in the cytosolic (global or local) ADP and ATP concentration or that there is an absolute concentration-dependence of these ligands on KATP channels. The relatively small conductance changes that have been found associated to active and silent phases (closing and opening of the KATP channels as interpreted by the authors, respectively, doi: 10.1152/ajpendo.00046.2013) and should be due to metabolic factors, could be also associated to desensitization of KATP channels to ATP due to the increase in cytosolic Ca2+ changes after intracellular Ca2+ flux (DOI: 10.1210/endo.143.2.8625) as they have been found to operate also at time scales, significantly faster (DOI: 10.2337/db22-0952) than reported before (refs. 21,22). Metabolic changes influence intracellular Ca2+ flux as well.

(4) There is no explanation for why KL divergence is so different between the pre-test regional consistency of the islets used to test the vehicle compared to those where GKa and PKa have been tested.

---

## [Author Response]

The following is the authors’ response to the original reviews.

**Public Reviews:**

**Reviewer #1 (Public Review):**
Summary:Jin, Briggs, and colleagues use light sheet imaging to reconstruct the islet threedimensional Ca2+ network. The authors find that early/late responding (leader) cells are dynamic over time, and located at the islet periphery. By contrast, highly connected or hub cells are stable and located toward the islet center. Suggesting that the two subpopulations are differentially regulated by fuel input, glucokinase activation only influences leader cell phenotype, whereas hubs remain stable.Strengths:The studies are novel in providing the first three-dimensional snapshot of the beta cell functional network, as well as determining the localization of some of the different subpopulations identified to date. The studies also provide some consensus as to the origin, stability, and role of such subpopulations in islet function.

We thank the reviewers for their positive assessment.

Weaknesses:Experiments with metabolic enzyme activators do not take into account the influence of cell viability on the observed Ca2+ network data. Limitations of the imaging approach used need to be recognized and evaluated/discussed.

We worked very hard to make sure the islets remained stable and healthy over the duration of imaging time course. We imaged the islet in 3D and observed that all betacells displayed glucose-dependent oscillations, which can only arise from functioning cells. From the raw calcium traces (displayed in the figures) we observed no detectable loss of signal over 60 min of continuous imaging regardless of drug treatment; this is because the laser excitation is below the bleach threshold for GCaMP6s, and it is bleaching that generates phototoxicity. To demonstrate this clearly, we performed a bleach test using 6x laser power; in this case calcium amplitude dropped 30% over a 60 min of imaging, however islet calcium oscillatory behavior was preserved. Light-sheet is well documented to be 1000x more gentle than other optical sectioning techniques, which is why it was chosen for this application.

Regarding the limitations of imaging approach, we recognized studying islets ex vivo is necessarily performed in the absence of native surrounding tissue, as highlighted in the discussion.

**Reviewer #2 (Public Review):**
The manuscript by Erli Jin, Jennifer Briggs et al. utilizes light sheet microscopy to image islet beta cell calcium oscillations in 3D and determine where beta cell populations are located that begin and coordinate glucose-stimulated calcium oscillations. The light sheet technique allowed clear 3D mapping of beta cell calcium responses to glucose, glucokinase activation, and pyruvate kinase activation. The manuscript finds that synchronized beta-cells are found at the islet center, that leader beta cells showing the first calcium responses are located on the islet periphery, that glucokinase activation helped maintain beta cells that lead calcium responses, and that pyruvate kinase activation primarily increases islet calcium oscillation frequency. The study is well-designed, contains a significant amount of high-quality data, and the conclusions are largely supported by the results.It has recently been shown that beta cells within islets containing intact vasculature (such as those in a pancreatic slice) show different calcium responses compared to isolated islets (such as that shown in PMID: 35559734). It would be important to include some discussion about the potential in vitro artifacts in calcium that arise following islet isolation (this could be included in the discussion about the limitations of the study).

Although isolated islets reproduce the slow oscillatory calcium behavior observed in vivo, we agree that missing elements such as blood flow, cholinergic innervation, and surrounding tissues may each impact islet calcium responses. Pancreatic regional blood flow also links the endocrine and exocrine signaling which can directly influence the behavior of beta cells. We have highlighted some of these issues in the discussion “In addition to α-cells, vasculature may also impact islet Ca2+ responses, and may induce additional heterogeneity in vivo.” (see line 375, Ref. 46).

**Reviewer #3 (Public Review):**
Summary:Jin, Briggs et al. made use of light-sheet 3D imaging and data analysis to assess the collective network activity in isolated mouse islets. The major advantage of using whole islet imaging, despite compromising on the speed of acquisition, is that it provides a complete description of the network, while 2D networks are only an approximation of the islet network. In static-incubation conditions, excluding the effects of perfusion, they assessed two subpopulations of beta cells and their spatial consistency and metabolic dependence.Strengths:The authors confirmed that coordinated Ca2+ oscillations are important for glycemic control. In addition, they definitively disproved the role of individual privileged cells, which were suggested to lead or coordinate Ca²⁺ oscillations. They provided evidence for differential regional stability, confirming the previously described stochastic nature of the beta cells that act as strongly connected hubs as well as beta cells in initiating regions (doi.org/10.1103/PhysRevLett.127.168101).The fact that islet cores contain beta cells that are more active and more coordinated has also been readily observed in high-frequency 2D recordings (e.g. DOI: 10.2337/db22-0952), suggesting that the high-speed capture of fast activity can partially compensate for incomplete topological information.They also found an increased metabolic sensitivity of mantle regions of an islet with a subpopulation of beta cells with a high probability of leading the islet activity which can be entrained by fuel input. They discuss a potential role of alpha/delta cell interaction, however relative lack of beta cells in the islet border region could also be a factor contributing to less connectivity and higher excitability.The Methods section contains a useful series of direct instructions on how to approach fast 3D imaging with currently available hardware and software.The Discussion is clear and includes most of the issues regarding the interpretation of the presented results.Some issues concerning inconsistencies between data presented and statements made as well as statistical analysis need to be addressed.Taken together it is a strong technical paper to demonstrate the stochasticity regarding the functions subpopulations of beta cells in the islets may have and how less well-resolved approaches (both missing spatial resolution as well as missing temporal resolution) led us to jump to unjustified conclusions regarding the fixed roles of individual beta cells within an islet.

We thank the reviewers for the comments on the many strengths of the manuscript and address the specific critiques below.

**Recommendations for the authors:**

**Reviewing Editor Comments:**
Essential revisions:(1) How useful is GK activation as a subpopulation-level perturbation, given that all beta cells would be affected? Previous studies by the authors have shown that GK gradients likely dictate subpopulation behaviour, so the concern here is that GK activation across all cells might mask the influence of such gradients i.e. a U-shaped effect. Also, does the GK activator differentially penetrate the islet such that first responders/leaders are more vulnerable than hubs?

As we previously published, non-saturating concentrations of GK activator (as used here) have the same effect on calcium oscillations as raising glucose (PMID:33147484). In other words, the activator boosts the activity of the endogenous GK. To the second point, recent ex vivo islet studies (PMID: 28380380) document the islet penetration of a fluorescent glucose analogue within seconds even under static conditions, and in our study the islets calcium oscillations reached steady state, so we are not concerned about drug penetration. The real limitation with any drug study in the islet is that non-beta cells are also activated; this limitation is included in the discussion along with the recommendation that genetic tools are needed to assess the effect of GK activation in the various endocrine subpopulations.

An additional concern with the GK activation experiment is that GK activation might push beta cells into a more stressed state such that they are more susceptible to phototoxicity. Although the authors state that photobleaching is low, they provide no data to support such a statement. Given the long duration of imaging and acquisition rate, phototoxicity might be more of an issue, especially with GK activation. Some further analysis (e.g. apoptosis) would be useful here to exclude an effect of beta cell viability versus GK activation on the observed phenotype of the different subpopulations.

Acute GK activation (for 30min) does not stress the islet; the drug has the same effect as raising glucose (PMID: 33147484). To determine whether photobleaching was impacted by GK activation, we examined the peak of consecutive oscillations in response to vehicle and GK activator. The average photobleaching was less than 2% of the calcium fluorescence over 30min of continuous imaging. Furthermore, GKa activation did not significantly increase photobleaching (see Author response image 1).

**Author response image 1. sa4fig1:** 

To the reviewer’s second point, apoptosis cannot occur on the timescale of the drug treatment (30min), and raw calcium traces are included showing that all beta cells display oscillatory behavior throughout the course of the experiment.

(2) The authors show that glucokinase activation increases the duration of islet calcium oscillations and in some islets (3 of 15 islets) causes "a Ca2+ plateau." The authors indicate that "Glucokinase, as the 'glucose sensor' for the β-cell, controls the input of glucose carbons into glycolysis, and opens KATP channels." It would be nice to have some experimental evidence that the change in oscillation rate caused by the glucokinase activator is due to KATP activation. This could be accomplished by treating islets with subthreshold KATP activators (e.g., diazoxide) or subthreshold KATP inhibitors (e.g., tolbutamide).

The statement that glucokinase activation opens KATP channels was a typo; glucose metabolism closes KATP channels by raising the ATP/ADP ratio. We now include additional citations that document the relationship between GK and KATP and the oscillatory behavior. See Ref 22 (PMID: 33147484) and Ref 34 (PMID: 33147484).

The manuscript finds that "Early phase cells were maintained to a greater degree upon GKa application." Yet GKa is proposed to activate KATP. Some discussion about how the early phase is maintained in cell populations by GKa activation in the context of KATP activity would be useful.

As discussed above, we meant to say that GKa will close KATP and apologize for the confusion. As we mentioned in the discussion, early phase cells are most likely maintained to a great degree following GK activation as result of enhanced GK gradient and reduced effect of stochastic alpha cell input.

(3) Membrane potential depolarization precedes calcium channel activation and subsequent calcium entry. In many cases, electrical coupling across beta cells happens on millisecond timescale. It would be good to confirm that the calcium is showing the same time scale in terms of elevation following beta cell membrane potential depolarization. One concern is that the islet beta cells could be depolarizing at the same speed and lagging in terms of calcium channel activation and calcium entry.

We thank the reviewer for making this point, which is almost certainly true, particularly since plasma membrane calcium influx is not the sole source of intracellular calcium. Previously published “simultaneous” recordings of Vm and calcium show their same phase relationship but do not have sufficient time resolution to capture depolarization of each cell. A quantification of phase lag would require the field to generate mice with voltage sensors expressed in beta cells; these tools are not yet available.

A related issue: in the text, the authors discuss changes in membrane potential (not been measured in this study), while in the figures they exclusively describe Ca2+ oscillations (which were measured). Examples are on lines 149, 150, 153, 154, 263. It is recommended that the silent and active phases in the Results section describe processes actually measured in this study as shown in 6A.

To clarify, we did not use the term ‘membrane potential’ anywhere in the manuscript. We do sometimes refer to calcium influx as a proxy for membrane depolarization; we think this is valid given the abundant evidence that these processes are interdependent in beta cells.

(4) It would be good to include the timing of the phases of calcium entry. When was the beta cell calcium entry monitored for the response time? Were the response times between the late and early phases consistent for each oscillation? It looks as if the start of the calcium upstroke was similar for many beta cells (such as for the Figure 2I traces). It would be nice to include a shorter time duration graph of calcium oscillation traces right when the upstroke starts. This would allow the community to observe the differences in the start time of calcium entry.

We agree this is an important point. We now include an inset showing the expanded time scale of the calcium upstroke in Fig.2I. The response time spread between early and late phase cells is now shown in Fig.7F (and in Author response image 2). We also quantified the coefficient of variation in the response time spread (0 = no variation and 1 = maximal variation) and found no significant differences between metabolic activators (Author response image 2).

**Author response image 2. sa4fig2:** 

Also, for most of the GCaMP6s traces shown, the authors indicate that they are plotted as F/F0. However, this normalization (F/F0) is not done for the actual traces shown. For example, Figure 2D shows the traces starting from what looks to be 0 to 0.3 F/F0, but the traces for an F/F0 group should all start at 1. Please change this for all representative oscillations so the start of calcium entry for example traces all line up.

This has been corrected in Fig. 2D, I and Fig. 3B. Also Fig.6 should be F not F/F0

**Reviewer #1 (Recommendations for the authors):**
(1) Line 53: "Silencing the electrical activity of these hub cells with optogenetics was found to abolish the coordination within that plane of the islet". The authors should acknowledge that studies also showed that beta cell transcription factor (Pdx1/Mafa) dosage was important for hub cell phenotype and islet function.

Thank you, this reference to Nasteska et al. (PMID: 33514698, Ref. 16) has been added to the discussion.

(2) Light sheet imaging is used to image the 3D islet volume. Whilst speed is undoubtedly an advantage of this technique, axial resolution is ~1.1 µm over 4 µm z-step size. How confident are the authors that single nuclei can be reliably identified given their ~6 µm size in a beta cell (e.g. do some elongated nuclear appear, which could be "doublets")?

The axial resolution of 1.1 µm exceeds the resolution needed for the Nyquist criterion (i.e. sampling every 2-3 µm). As a practical matter, it is not possible to doublecount nuclei because the software will exclude nuclei that occupy the same volume. Only a very elongated nucleus (>10 µm) would be double counted and this does not occur.

(3) The authors discuss the advantages of the light sheet imaging approach used, including speed and phototoxicity. Some more balance is needed here since other approaches such as two-photon excitation achieve similar speeds with much better axial resolution (see dozens of neural circuit studies).

We are careful to point out that two-photon excitation has better axial resolution, better tissue penetration, and often higher speeds (kHz using linescans) – however these neuronal studies are limited to the cells in a few planes and the laser power is orders of magnitude higher than lightsheet. For this reason, two photon imaging has not been used to image islet calcium in three dimensions. The bottom line is lightsheet trades axial resolution for gentle volumetric imaging.

(4) Line 340: "Laser ablation or optogenetic inactivation of these early phase cells would be predicted to have little impact on islet function, as suggested previously by electrophysiological studies in which surface β-cells have been voltage-clamped with no impact on β-cell oscillations". This statement is slightly ambiguous since the authors showed in their previous studies that laser ablation of first responder cells/leaders was able to influence the Ca2+ network. Do the authors mean that laser ablation would only temporarily influence islet function before another cell picked up the role of a first responder/leader? As written, the sentence seems to imply that first responders/leaders are unimportant for the islet function.

We intended to imply that the oscillatory system is sufficiently robust that a new cell take over when leader cells are ablated. We also cite Korosak et al. (PMID:34723613, Ref. 40) and Dwulet et al. (PMID: 33939712, Ref. 15) to make this point, although to clarify we are not examining first responders in this study.

(5) Line 369: "In contrast with leader cells, we found that the highly synchronized cells are both spatially and temporally stable." The sentence needs qualifying- what would spatiotemporal stability be expected to confer on such a subpopulation?

We believe that the spatiotemporal stability of highly synchronized cells is a consequence of beta cells in the center of the islet lacking the stochastic input of nearby alpha cells; we raise this point in the discussion: “The preponderance of α-cells on the periphery of mouse islets, which influence β-cell oscillation frequency, would be expected to disrupt β-cell synchronization on the periphery and stabilize it in the islet center – which is precisely the pattern of network activity we observed.” (see line 372).

(6) Line 370: "However, in conflict with the description of hub cells as intermingled with other cells throughout the islet, the location of such cells in 3D space is close to the center." The study by Johnston et al did not have the axial resolution to exclude that some cells might have been grouped together.

We agree and have included the reviewer’s comment in the text (See line 384); that’s an important reason for conducting this 3D study.

(7) Line 380: "One explanation may be that paracrine communication within the islet determines which region of cells will show high or low degree. For example, more peripheral cells that are in contact with nearby δ-cells may show some suppression in their Ca2+ dynamics, and thus reduced synchronization." A potentially exciting future study. Should however probably cite DOI s41467-022-31373-6 here.

We thank the reviewer for their input. This reference to Ren et al. (PMID:35764654) was previously included as Ref. 42 (now Ref. 45)

**Reviewer #3 (Recommendations for the authors):**
(1) There are in fact no radially oriented networks in the core of an islet (l. 130, Figure 4) apart from the fact that every hub has somewhat radially oriented edges. For radiality to have some general meaning, the normalized distance from the geometric center would need to be lower than 0.4. The networks are centrally located, which does not change the major conclusions of the study.

Thank you for pointing out this imprecise language. We did not intend to imply that the functional network is orientated radially. We corrected the text (see line 131, 145) to indicate that the cells with high and low synchronization are distributed in a radial pattern.

(2) The study would benefit from acknowledging that Ca2+ influx is not a sole mechanism to drive insulin secretion and that KATP channels are not the sole target sensitive to changes in the cytosolic (global or local) ADP and ATP concentration or that there is an absolute concentration-dependence of these ligands on KATP channels. The relatively small conductance changes that have been found to be associated with active and silent phases (closing and opening of the KATP channels as interpreted by the authors, respectively, doi: 10.1152/ajpendo.00046.2013) and should be due to metabolic factors, could be also associated to desensitization of KATP channels to ATP due to the increase in cytosolic Ca2+ changes after intracellular Ca2+ flux (DOI: 10.1210/endo.143.2.8625) as they have been found to operate also at time scales, significantly faster (DOI: 10.2337/db22-0952) than reported before (refs. 21,22). Metabolic changes influence intracellular Ca2+ flux as well.

The reviewer is absolutely correct that there are amplifying factors and other sources of calcium beyond plasma membrane influx and there are other mechanisms that regulate insulin secretion beyond calcium levels. These alternative mechanisms are introduced in Refs. 1-2, however they are not the focus of this study.

(3) There is no explanation for why KL divergence is so different between the pre-test regional consistency of the islets used to test the vehicle compared to those where GKa and PKa have been tested.

We thank the reviewer for their careful observation. This arises because there are larger differences between preparations than within a preparation. This has been described previously (PMID: 16306370 and 20037650) and could be expected to account for the differences in KL divergence between animals.

(4) Statistical analysis would profit from testing the normality of the data distribution before choosing the statistical test and then learning the difference between parametric and nonparametric tests. For example, in Figures 3CD and 5EF, the data density is lower at the calculated mean than below and above this value and there are other examples in other figures too.

We thank the reviewer for this very important comment, and we apologize for the oversight on our part. To address this comment, we conducted two normality tests: Anderson-Darling and Kolmogorov-Smirnov on all statistical analyses in the manuscript. If the data were not normally distributed, we changed the analysis to Wilcoxon matchedpairs signed rank test (non-parametric version of t-tests) or the Friedman test (nonparametric version of ANOVA). Three results were changed based on this statistical correction: Figure 4D, also 5F 3D (from P=0.01 to P=0.0526), Figure 5F ¼ z-depth (P = 0.005 to P = 0.012). We have updated the manuscript methods, results, and figures accordingly. Importantly, these results did not change the main points of the paper.